# Novel Approaches to Treatment of Acute Myeloid Leukemia Relapse Post Allogeneic Stem Cell Transplantation

**DOI:** 10.3390/ijms241915019

**Published:** 2023-10-09

**Authors:** Carmine Liberatore, Mauro Di Ianni

**Affiliations:** 1Hematology Unit, Department of Oncology and Hematology, Santo Spirito Hospital, 65124 Pescara, Italy; carmine.liberatore@asl.pe.it; 2Department of Medicine and Sciences of Aging, “G. d’Annunzio” University of Chieti-Pescara, 66100 Chieti, Italy

**Keywords:** acute myeloid leukemia, relapse, allogeneic stem cell transplantation

## Abstract

The management of patients with acute myeloid leukemia (AML) relapsed post allogeneic hematopoietic stem cell transplantation (HSCT) remains a clinical challenge. Intensive treatment approaches are limited by severe toxicities in the early post-transplantation period. Therefore, hypomethylating agents (HMAs) have become the standard therapeutic approach due to favorable tolerability. Moreover, HMAs serve as a backbone for additional anti-leukemic agents. Despite discordant results, the addition of donor lymphocytes infusions (DLI) generally granted improved outcomes with manageable GvHD incidence. The recent introduction of novel targeted drugs in AML gives the opportunity to add a third element to salvage regimens. Those patients harboring targetable mutations might benefit from IDH1/2 inhibitors Ivosidenib and Enasidenib as well as FLT3 inhibitors Sorafenib and Gilteritinib in combination with HMA and DLI. Conversely, patients lacking targetable mutations actually benefit from the addition of Venetoclax. A second HSCT remains a valid option, especially for fit patients and for those who achieve a complete disease response with salvage regimens. Overall, across studies, higher response rates and longer survival were observed in cases of pre-emptive intervention for molecular relapse. Future perspectives currently rely on the development of adoptive immunotherapeutic strategies mainly represented by CAR-T cells.

## 1. Introduction

Allogeneic hematopoietic stem cell transplantation (HSCT) currently represents the only potentially curative approach for many patients with acute myeloid leukemia (AML) and high-risk myelodysplastic syndromes (MDSs) [1]. The availability and outcome of HSCT have progressively improved over the last decades due to advances in donor selection, transplantation techniques, and supportive care [2,3]. However, the incidence of disease relapse has largely remained unchanged, and, as a consequence, it has become the major cause of treatment failure [4]. Overall, the relapse of AML and MDS involves about 30% of patients undergoing HSCT, although rates up to 80% are reported depending on patient, disease, and transplantation characteristics [4,5,6,7,8]. The relapse rate varies depending on the type of donor: among patients with AML in first complete remission, the incidence of relapse at 3 years in matched sibling donors (MSDs), matched unrelated donors (MUDs), and haploidentical donors ranges between 14 and 30%, 20 and 31%, and 14 and 38%, respectively [9,10,11]. Actually, the use of haploidentical HSCT has progressively increased in recent years, and the use of post-transplantation cyclophosphamide (PTCy) has become the most common platform for GvHD prophylaxis [12]. However, haploidentical HSCT based on T regulatory cells (Tregs)/T conventional cells (Tcons) immunotherapy, despite being more sophisticated, has shown unprecedently low rates of relapse [13]. Relapsed patients generally have a poor prognosis. After different conventional salvage treatments, such as low-dose or intensive chemotherapy (IC), donor lymphocytes infusions (DLIs), and a second HSCT, only about 30% of patients achieve complete response (CR) whereas long-term outcomes are obtained in fewer than 20% of them [4,5,6,7,8]. Notably, long-term survival largely relies on the successful induction of CR by cytoreductive therapy followed by an immunological consolidation with either DLI or subsequent HSCT [6,7,8,14]. Unfortunately, in the early post-transplantation period, the severe toxicities related to IC as well as a second HSCT could represent a major limit for less fit and heavily pre-treated patients. Hence, there is a need for novel targeted therapeutic approaches able to overcome mechanisms of relapse and exploit the graft-versus-leukemia effect (GvL) with tolerable toxicities and limited graft-versus-host disease (GvHD) (Figure 1).

## 2. Donor Lymphocytes Infusions

The infusion of peripheral lymphocytes from a donor is one of the first and simplest forms of post-transplantation immunotherapy. Despite impressive results in chronic myeloid leukemia relapsing after HSCT, this procedure appeared less successful in treating a rapidly proliferative disease, such as AML [15]. However, DLIs are still largely used in this setting, either alone or in combination with additional antileukemic or immunomodulating agents. In the presence of measurable residual disease (MRD) positivity and/or increasing mixed chimerism, a growing body of evidence suggests that pre-emptive DLIs are a safe and effective method for preventing hematological relapse [16]. A retrospective analysis on pre-emptive DLI in a group of 80 patients with MRD-positive acute leukemias after HSCT showed a cumulative incidence of 6% in patients treated with DLI compared to 63% without DLI [17]. The protective effect of pre-emptive DLI was confirmed in two prospective trials where administration of DLI was associated with a reduction in relapse risk [18,19].

In the context of overt hematological relapses, the clinical benefit of DLI is limited to a minority of patients. A retrospective study by EBMT analyzed the outcomes of 399 patients with AML in first hematological relapse after HSCT whose treatment did (n = 171) or did not (n = 228) include DLI [14]. The estimated overall survival (OS) at 2 years was 21% for patients receiving DLI and 9% for patients not receiving DLI, and in the multivariate analysis, the use of DLI was significantly associated with longer survival. Among DLI recipients, lower tumor burden at relapse, favorable cytogenetics, and remission at time of DLI were predictive for survival. The incidence of acute and chronic GvHD was 43% and 46%, respectively. Notably, the onset of chronic GvHD was associated with a better outcome [14].

A particular condition for the administration of DLI is represented by haploidentical HSCT, where the high donor–recipient disparity could lead to an increased risk of severe GvHD. Furthermore, in haploidentical HSCT, it is also recommended to determine the presence of HLA-loss variants that render leukemic cells invisible to alloreactive T cells but leave the risk of GvHD unchanged [20]. In the first study reporting on the use of DLI for relapses after haploidentical HSCT, the 2-year leukemia-free survival was 40%, although it was accompanied by significant risks of both severe acute GvHD (30%) and chronic GvHD (64%). Subsequently, two studies testing DLI in a dose-escalation manner after PTCy-based haploidentical HSCT reported CR rates of 30–45% but a lower risk of both acute and chronic GvHD [21,22]. Among recipients of haploidentical HSCT from G-CSF mobilized peripheral stem cells and PTCy, the administration of DLI in the case of disease relapse or loss of chimerism obtained similar results [23,24]. A large retrospective study on behalf of EBMT analyzed the outcomes of 173 patients with hematological malignancies (80 patients with AML) treated with DLI after haploidentical HSCT and PTCy-based GvHD prophylaxis. Indications for DLI were either relapse prevention in high-risk patients or treatment of both molecular and hematological relapse. In the case of pre-emptive DLI, the median number of infusions was two, with a median first dose of 0.5 × 106 CD3+ T cell/kg, whereas the median number of DLIs was one, with 1 × 106 CD3+ T cell/kg in the case of hematological relapse. Despite the heterogeneity of populations and treatments, the outcomes appeared better when DLIs were given for early interventions: the 2-year OS was 40% for pre-emptive and 22% for therapeutic DLI. The incidence of grade II–IV acute GvHD and chronic GvHD ranged between 20 and 21% and 17 and 24%, respectively [25]. In patients relapsing after haploidentical HSCT with Tregs/Tcons platform, a case report showed that Tregs-protected DLIs exert a strong GvL effect with limited GvHD [26].

## 3. Hypomethylating Agents

The hypomethylating agents (HMAs) Azacitidine (AZA) and Decitabine (DEC) have been increasingly used for the treatment of AML and high-risk MDS relapsed after HSCT. The major advantage of HMA over IC is a lower toxicity profile, whereas the rationale for their use relies on both anti-leukemic and immunomodulating effects [27]. In preclinical models, HMAs have been shown to increase the responses of tumor-specific CD8+ T cells through the upregulation of several epigenetically silenced tumor antigens and HLA and costimulatory molecules on the leukemic cell surface [28,29,30,31,32,33,34,35]. AZA also favored the upregulation of inhibitory pathways and the expansion of regulatory T cells [36,37,38,39]. Despite potentially attenuating T cells’ alloreactivity, these immunoregulatory properties of HMA might in part prevent or reduce GvHD [40,41].

Azacitidine, either as a single agent or in combination with DLI, has been used at different dosages and schedules in patients with both molecular and hematologic relapse of AML and MDS after HSCT. The most relevant retrospective studies and five non-randomized prospective trials showed efficacy and good tolerability [24,42,43,44,45,46,47,48,49,50]. The high heterogeneity of reports accounted for response rates ranging between 10 and 49% in the case of hematologic relapses and between 33 and 80% when AZA was given as pre-emptive therapy. Similarly, the OS at 2 years ranged between 12% and 54% in the first group and between 25% and 70% in the latter group (Table 1) [24,42,43,44,45,46,47,48,49]. Altogether, AZA proved much more effective when employed for the treatment of molecular relapse. Lower disease burden and a longer interval between HSCT and disease relapse generally predicted a better response and significant survival advantage, whereas the benefit of combination with DLI was less clear across studies [44,46,49,50]. An attempt to identify mutations predictive of a response and outcome in patients receiving AZA after HSCT was performed by Woo et al. Although the number of patients was too small to draw definitive conclusions, the *TP53* mutation was significantly associated with a poor response to AZA and inferior survival. Conversely, a trend toward better outcomes emerged in patients with *TET2* mutation [47]. Besides the expected hematological toxicities of AZA, no relevant extra-hematological adverse events or GvHD exacerbation were observed. The safety of AZA in combination with DLI was also proved in the case of patients relapsing after HSCT from alternative donors, where the HLA mismatches could represent a matter of concern for severe GvHD [24].

The use of Decitabine for the treatment of myeloid neoplasms relapsed after HSCT is more limited. Prospective trials are lacking; indeed, results from retrospective case series are similar to those reported with AZA in terms of efficacy and the safety profile (Table 1) [51,52,53]. To date, the major experience comes from a retrospective study of the German Cooperative Transplant Study group. Among 36 patients who received DEC either as a single agent or in combination with DLI (n = 22, 61%), the overall response rate (ORR) was 25%, including 6 patients with CR (17%), and the 2-year OS rate was 11% [53]

Besides their use as a single agent or in combination with DLI, HMAs could also serve as a backbone for combination treatment with novel targeted agents.

## 4. Lenalidomide

Lenalidomide has shown promising anti-leukemia activity, although it is burdened by severe GvHD when employed in post-transplantation settings [54]. Therefore, the rationale to combine Lenalidomide with AZA relies on the synergism of cytotoxic effects as well as the attempt to mitigate GvHD exacerbation thanks to the immunomodulating properties of AZA. The prospective phase I VIOLA trial investigated a salvage regimen with standard dosage AZA followed by escalating doses of Lenalidomide up to 25 mg daily in 29 patients with hematological relapse of AML and MDS post HSCT. Treatment was well tolerated, with limited GvHD incidence. After a median of three cycles, the ORR and CR were 47% and 20%, respectively, and the median OS was 27 months among responders [55]. Recently, the AZALENA trial confirmed the tolerability and efficacy of the combination of AZA and low-dose Lenalidomide as a first salvage therapy in 50 patients with post-HSCT relapse of AML and MDS. Both molecular (52%) and hematological (48%) relapse were included. Notably, as per the protocol, 34 patients (68%) also received DLI at escalating dosages. After a median of seven cycles, the ORR and CR were 56% and 50%, respectively. After a median follow-up of 20 months, the median OS was 21 months, whereas the median OS was not reached in those patients who achieved CR. Treatment was well tolerated without an excess of GvHD or toxicity (Table 2) [56]. Notably, both studies analyzed the circulating lymphocytes of patients before and after treatment, demonstrating impaired CD8+ T cells activity and a high incidence of CD4+ FOXP3+ regulatory T cells at baseline. These features were consistent with T cells exhaustion and an immunosuppressed microenvironment, and neither was reversed during treatment [55,56].

## 5. Venetoclax

The anti-apoptotic BCL2 inhibitor Venetoclax in combination with either HMA or chemotherapy has shown unprecedent efficacy in AML, and it is currently approved for the treatment of elderly patients ineligible for IC [73,74]. Promising results have also been obtained in patients with relapsed/refractory disease [75]. Subsequently, Venetoclax has been employed in the post-transplantation setting.

A prospective phase I/II trial combined Venetoclax with the FLAG-Ida chemotherapeutic regimen in newly diagnosed and relapsed/refractory AML. Among the 68 patients enrolled, 14 had previously received HSCT. After a median of two cycles, the ORR and composite CR were 70% and 57%, respectively, while the median OS in the phase II cohort was not reached after a median follow-up of 12 months. The main grade III–IV adverse events were both hematological and infectious, occurring at up to 90% in the post-transplantation setting (Table 2) [76].

In combination with HMA, available data come from several retrospective single center experiences and reports from collaborative groups. Starting with a short ramp-up, Venetoclax was given continuously in addition to either AZA or DEC at standard dosages. The majority of patients had overt hematologic relapse and advanced disease stage. Altogether, the response rates ranged between 38% and 64%, whereas the median OS ranged between 2 and 10 months [57,58,59,61,62,63]. Despite disparities due to the great heterogeneity of the included patients, the results appeared slightly better than those obtained with HMA monotherapy. Moreover, an attempt to predict outcomes with biological features showed that patients harboring *TP53* mutations did not benefit from Venetoclax-HMA, whereas *IDH*-mutated AML showed higher response rates [58,60]. Otherwise, Schuler et al. observed better outcomes in those patients who received Venetoclax-HMA as the first salvage treatment and among those treated in molecular relapse [62]. Among patients relapsed after HSCT, DLIs were combined with Venetoclax and HMA in a limited number of studies. Along with favorable responses, a low incidence of GvHD as well as good tolerance of treatment were observed [61,62,63]. In a retrospective multicenter trial, Venetoclax monotherapy and DLI at escalating doses were given to 22 patients with early relapse of AML after HSCT. Acute and chronic GvHD were observed in four (18%) and six (27%) patients, respectively. After a median of two cycles of Venetoclax and one DLI, the ORR was 50% and the median OS was 6.1 months (Table 2) [77].

Besides efficacy, the addition of Venetoclax to HMA was burdened by severe hematological and infectious complications across reported studies, thus generally limiting the median number of administered cycles to two per patient. Grade III–IV neutropenia and thrombocytopenia are reported in 70–90% of patients, leading to frequent treatment interruptions and dose reductions in order to allow hematological recovery [60,61,62,63]. Despite both antimicrobial and antifungal prophylaxis, grade III–IV infections occurred in 50–70% of cases, including invasive fungal infections and fatal complications [59,60,62,63]. Indeed, these results pose a problem of tolerability when Venetoclax is employed in more fragile and immunosuppressed patients, such as those relapsed after HSCT. Therefore, prospective trials are warranted to accurately investigate the dosing and schedule of Venetoclax post HSCT as well as its use in the earlier setting of molecular relapse.

The hyperexpression of pro-survival and anti-apoptotic protein MCL1 is a known mechanism of resistance to Venetoclax in AML [78]. Preclinical models showed that Actinomycin D was able to overcome resistance to Venetoclax through inhibition of MCL1 [79,80]. Actually, in a single-center retrospective experience, Zucenka et al. demonstrated the clinical benefits of the Venetoclax, low-dose cytarabine, and Actinomycin D regimen (ACTIVE) in 20 patients with AML relapsed after HSCT over a historical cohort of 29 patients with similar characteristics treated with FLAG-Ida. Responding patients could also receive maintenance with Venetoclax, low-dose cytarabine, and DLI. The ACTIVE regimen granted a higher ORR (75% vs. 66%; *p* = 0.542) and CR (70% vs. 34%; *p* = 0.002) compared to FLAG-Ida and was well tolerated, with a shorter time to hematological recovery and lower infectious complications; both translated into a significantly lower treatment-related mortality (0% ACTIVE vs. 34% FLAG-Ida; *p* = 0.003). The median OS was longer compared to FLAG-Ida patients (13.1 vs. 5.1 months, *p* = 0.032), and the survival advantage was maintained even after censoring patients at the time of the second HSCT [64]. Homoharringtonine is an herbal compound with known antileukemic properties that proved to synergistically enhance the activity of Venetoclax and overcome resistance by reducing MCL1 expression and inducing apoptosis in myeloid leukemia cells [81,82]. Recently, in a prospective multicenter phase II clinical trial, the salvage regimen of Venetoclax, AZA, and Homoharringtonine (VAH) in 96 patients with relapsed/refractory AML (43 post HSCT) granted high response rates (ORR 78.1% and CR 70.8%) and prolonged survival (median OS 22.1 months) with an acceptable safety profile, paving the way to further combinational therapies of Venetoclax with MCL1 inhibitors [65,79].

## 6. IDH1/IDH2 Inhibitors

Patients with AML present recurrent mutations in isocitrate dehydrogenase (IDH)1 and IDH2 genes in about 10% of cases [83]. Ivosidenib and Enasidenib are two orally available selective inhibitors of IDH1- and IDH2-mutated proteins that showed efficacy in the treatment of relapsed/refractory AML. To date, the only data available in the post-transplantation setting come from the results of the pivotal trials in which some patients who relapsed after HSCT were included.

In a prospective phase I/II clinical trial, Enasidenib was given to 239 patients with relapsed/refractory AML. In total, 24 patients out of 176 in the efficacy cohort had relapsed after HSCT. After a median of five cycles at the recommended dose of 100 mg once daily, the ORR and CR were 40.3% and 19.3%, respectively. The median OS was 9.3 months in the whole cohort, whereas the median OS reached 19.7 months among responders. Grade III–IV adverse events were limited to hyperbilirubinemia (12%) and differentiation syndrome (7%). Due to the cellular differentiation and maturation effects, responses to Enasidenib occurred in the absence of bone marrow aplasia, with limited hematological toxicities and infectious complications [84]. Similar results were reported in 258 patients with relapsed/refractory AML treated with Ivosidenib. Among 125 patients in the primary efficacy cohort, 36 had received HSCT. The ORR and composite CR were 41.6% and 30.4%, respectively. The median duration of the CR and the median OS were 8.2 months and 8.8 months, respectively. The safety profile was favorable, as grade III–IV adverse events were mainly QT interval prolongation (7.8%) and differentiation syndrome (3.9%) [85].

In the post-HSCT setting, both Enasidenib and Ivosidenib have recently been applied as maintenance therapy. In a multicenter phase I trial, 23 patients with IDH2-mutated AML and high-risk MDS received Enasidenib starting between 30 and 90 days following HSCT for 12 cycles. At the recommended dosage of 100 mg daily, adverse events and GvHD were limited. At 2 years, the cumulative incidence of relapse was 16%, whereas the PFS and OS were 69% and 74%, respectively [86]. A similar trial with Ivosidenib 500 mg as maintenance following HSCT proved equally tolerable with same efficacy results [87]. Nevertheless, considering the small sample size of these trials, larger prospective studies are warranted to assess the true potential of IDH1/2 inhibitors after HSCT.

## 7. FLT3 Inhibitors

Patients with FLT3-mutated AML may benefit from tyrosine kinase inhibitors (TKI) in the management of disease relapse. Besides direct antileukemic activity, the first-in-class multi-kinase inhibitor Sorafenib has shown immunomodulating properties in the context of allogeneic transplantation by promoting GvL activity through IL-15 production in FLT3-internal-tande-duplication (FLT3-ITD) mutated leukemic cells [88]. Alloimmune properties were indirectly confirmed in a retrospective study by Metzelder et al. where relapsed/refractory FLT3-ITD-mutated AML patients were treated with Sorafenib monotherapy. Among patients relapsed after HSCT, both higher response rates and longer survival were observed compared to patients who did not receive HSCT. Additional DLIs were given to 10 out of 29 patients, without relevant GvHD exacerbation [66]. Notably, in the long-term follow-up analysis of the transplantation cohort, 6 out of 29 patients (21%) were still alive after a median follow-up of 7.5 years. The achievement of a deep and stable molecular response appeared as a strong predictor of long-term survival. Besides one patient who received a subsequent HSCT, the remaining five patients maintained a stable CR with Sorafenib monotherapy, and four of them could stop treatment, suggesting a curative potential of Sorafenib [89]. More recently, a large retrospective study included 53 patients with FLT3-ITD-mutated AML relapsed after HSCT and treated with Sorafenib either as monotherapy or in combination with chemotherapy and/or DLI. Altogether, the ORR and CR were 83% and 66%, respectively, with a median duration of CR of 152 days. At 1 year, the OS was 46.8% and the PFS was 44.9%. Sorafenib was well tolerated, and major adverse events were represented by hematological toxicities, increased liver enzymes, and typical foot–hand syndrome. Cumulative incidences of acute and chronic GvHD were 32.0% and 28.1%, respectively. Notably, the addition of DLIs did not increase the occurrence of GvHD, whereas they appeared as the only predictive factors for longer survival in multivariate analysis [67]. Following the results of a phase II study exploring the clinical activity of Sorafenib with AZA [90], this combination has also been utilized in the post-HSCT setting. To date, the results of two retrospective case series are available; they report a CR ranging from 40 to 60%, a median OS of about 11 months, and an acceptable safety profile (Table 2) [68,69,70].

The experience with second-generation TKI is more limited in case patients with prior HSCT. Similar to Sorafenib, a preclinical study showed that Gilteritinib was able to enhance GvL’s effect by increasing IL-15 expression in FLT3-ITD-mutated leukemic cells and through down-regulation of co-inhibitory receptors, such as PD-1 and TIGIT, on donor CD8+ T cells [91]. In the phase III ADMIRAL trial that demonstrated the advantage of Gilteritinib over conventional salvage regimens for relapsed/refractory FLT3-mutated AML, 48 out of 247 treated patients had received prior HSCT. In this cohort, Gilteritinib obtained a CR of 35.4% and a median OS of 8.3 months, with a similar safety profile [71].

Preclinical models of FLT3-mutated AML showed synergistic anti-leukemic activity of Gilteritinib when combined with Venetoclax [92,93]. Then, a recent phase I open-label trial enrolled 65 patients with relapsed/refractory AML to receive Gilteritinib 80–120 mg and Venetoclax 400 mg once daily. In total, 36 out of 56 (64%) FLT3-mutated AML patients had already received prior TKI, and 19 patients (31%) had received prior HSCT. As expected, the most common grade III–IV adverse events were cytopenias (80%), leading to temporary treatment interruptions in about 50% of patients to allow hematological recovery. The ORR and composite CR were 75% and 39%, respectively, without differences among patients previously exposed or not to FLT3 inhibitors. Responses were also rapid (median time to response 0.9 months), deep (MRD negativity was achieved in 60% of evaluable patients), and durable (median CR duration 4.9 months). After a median follow-up of 17.5 months, the median OS for FLT3-mutated patients was 10.0 months [72]. Further results and prospective trials are awaited considering the limited data currently available on the clinical activity of Gilteritinib in the post-transplantation setting.

## 8. Checkpoint Inhibitors

The loss of donor T cells’ alloreactivity either by progressive CD8+ T cell dysfunction or overexpression of inhibitory immune-checkpoint molecules on leukemic cells’ surface is the basis of disease relapse after HSCT [94]. Therefore, immune checkpoint inhibitors have been employed in this setting with the purpose of restoring the GvL effect [95].

Whereas the blockade of the PD-1/PD-L1 pathway was burdened by severe GvHD exacerbation, in murine models of disease relapse after transplantation, the selective inhibition of cytotoxic T lymphocyte–associated protein 4 (CTLA-4) increased the GvL effect with limited immune-related toxicities [96]. In a phase I multicenter trial, Ipilimumab was tested on 28 patients with late relapse of hematological malignancies post HSCT (12 patients had AML and 2 patients had MDS). Ipilimumab confirmed its favorable safety profile, although immune-related adverse events and clinically relevant GvHD were observed in 21% and 14% of patients, respectively. Among 20 patients treated at the recommended dose of 10 mg/kg, the ORR and CR were 32% and 23%, respectively, including 4 patients with extramedullary AML that obtained CR. The 1-year OS was 49% in the whole cohort. Notably, clinical responses correlated with a decrease in circulating CD4+ regulatory T cells and an increase in CD4+ effector T cells, as well as with the infiltration of the tumor microenvironment by cytotoxic CD8+ T cells [97]. In the attempt to augment efficacy and limit adverse events, Ipilimumab has been combined with DEC in a multicenter phase I trial that enrolled relapsed/refractory AML and MDS patients. Among 25 patients in the post-HSCT cohort, immune-related adverse events occurred in 44% of patients and were dose dependent. Unfortunately, in these patients with earlier relapse from transplantation (<12 months), the efficacy of combination therapy mainly relied on the cytotoxic properties of DEC and generally appeared modest, with an ORR of 20% and a median duration of response of 4.46 months [98,99].

The PD-1 inhibitor Nivolumab showed a lack of efficacy coupled with relevant toxicity in terms of GvHD incidence when utilized in a small cohort of 10 patients with AML relapsed post HSCT [100]. Therefore, Nivolumab was combined with AZA with the purpose of overcoming the immunological resistance to HMA and to mitigate toxicities of the PD-1/PD-L1 blockade. Seventy patients with relapsed/refractory AML were treated in a phase II trial. The salvage regimen was safe and tolerable: the incidence of grade III–IV immune-related adverse events was 11%, but only 13 enrolled patients had prior HSCT. The overall response rates and CR were 33% and 22%, respectively, whereas the median OS was 6.3 months [101]. To date, despite encouraging results in the extramedullary relapse of AML treated with Ipilimumab, the efficacy of checkpoint inhibitors still appears modest in treating AML, whereas immune-mediated toxicities as well as the risk of GvHD remain an open issue in the post-transplantation setting.

## 9. Second Allogeneic Stem Cell Transplantation

A second HSCT is another possible choice for the management of AML relapsing after a first HSCT. However, a second HSCT is burdened with severe toxicities and, as a consequence, few fit patients are eligible, especially in the early post-transplantation period. Schmid et al. reported that only a minority of patients (15%) relapsing after the first HSCT could undergo a second transplantation, with response rates ranging between 41% and 56% and a 2-year OS between 15% and 42% [6]. Similarly, in a study by CIBMTR, a second HSCT was possible in only 369 patients (21%), with conditioning regimens being myeloablative in 49% and reduced-intensity/non-myeloablative in 30% of patients [7]. Although only a minority of relapsed patients could afford it, the second HSCT resulted in a higher response rate (CR 44%) and longer survival than other salvage treatments [7]. The use of a reduced-intensity conditioning regimen has progressively increased over the last years, reaching 77% of patients in a recent large registry analysis [8].

To date, limited data exist to support the use of a different donor for the second HSCT in the attempt to induce a stronger GvL. To address the role of donor change for acute leukemia relapsing after a first HSCT, the German transplantation group performed a retrospective registry study on 179 s HSCT (MSD, n = 75; MUD, n = 104) using either the same or a different donor. The outcome of the second HSCT generally appeared better after a first HSCT from MSD than from MUD (2-year OS: 37% vs. 16%, respectively; HR, 0.68; 95% CI, 0.47 to 0.98; *p* = 0.042). Interestingly, selecting a new donor for the second HSCT did not significantly improve the OS [102]. A subsequent EBMT analysis also demonstrated no difference in the cumulative incidence of relapse, non-relapse mortality, or OS when the same (n = 1884) or a different donor (n = 712) was chosen for the second HSCT [103]. In the setting of haploidentical HSCT, a retrospective analysis including 556 patients relapsing after the first HSCT found no differences in terms of response rates or survival between patients receiving from the same donors, a different matched donor, and a haploidentical donor for the second HSCT. Leukemia-free survival at 2 years was 23.5%, 23.7%, and 21.8%, respectively (*p* = 0.30), although a haploidentical donor predicted higher non-relapse mortality [104]. Overall, donor change was not detrimental in either of these studies and has become a common clinical practice. Indeed, in a large registry analysis of 8162 AML patients relapsing after transplantation between 2000 and 2018, Bazarbachi et al. reported a sharp increase over time in the use of a different donor from 31% in 2000–2004 to 80% for 2015–2018, respectively (*p* = 0.001) [8]. Additionally, improved survival has been observed in recent years, including those patients with early relapse, with an OS exceeding 30% among both younger and elderly patients [8]. Multivariate analysis in several studies found that an extended disease control may be obtained in those patients with good performance status, complete remission at the time of the second HSCT, and, most importantly, longer remission duration after the first HSCT [8,102].

## 10. Chimeric Antigen Receptor T Cells

The radical revolution that CAR-T cells are bringing in the treatment of lymphoid and plasma cell malignancies is slowly involving even the field of myeloid neoplasms. The main obstacle to overcome when treating AML still remains the identification of a specific target on the leukemic cell surface. Actually, suitable antigens for the development of anti-AML CAR-T cells, such as CD33 and CD123, are also commonly expressed on normal hematopoietic stem cells and myeloid progenitors as well as non-hematopoietic cells, thus predisposing them to on-target off-tumor adverse events, such as multilineage cytopenias [105,106]. Moreover, prolonged post-infusion cytopenias appeared as a common off-target off-tumor adverse event, even in patients treated with CAR-T cells for lymphoid and plasma cell neoplasms, possibly limiting the tolerability of these treatments in the post-HSCT setting [107]. Recently, the surface receptor CD44v6 prompted attention due to its selective expression on AML leukemic cells, and, in preclinical models, CD44v6-directed CAR-T cells showed consistent efficacy against AML while sparing normal tissues [108].

To date, reports including patients relapsed after HSCT are available on CAR-T cells directed against myeloid antigens, such as CD33, CD38, and CD123, as well as CD19, which is aberrantly expressed in AML with t(8;21) [109,110,111,112,113]. A recent meta-analysis including 57 patients reported a pooled ORR and CR of 65.2% and 49.5%, respectively, whereas the duration of response and OS were highly variable [114]. The infusion of CAR-T cells post HSCT proved safe across studies, but efficacy remains modest, thus highlighting the need for further prospective trials with a greater number of patients and longer follow-up. Notably, the major concern with CAR-T cells therapy in the post-HSCT setting is represented by the risk of GvHD due to the potential alloreactivity of T cells [115,116]. Different approaches to mitigate GvHD include the use of allogeneic virus-specific T cells, TCR-deficient T cells, and regulatory T cells [117]. Moreover, the insertion of a suicide gene might represent a strategy to manage GvHD flares and severe toxicities by enabling fast ablation of CAR-T cells [108]. Another possibility to mitigate GvHD in the post-transplantation setting is to generate CAR-T cells from the recipient’s T cells. Despite the possibility of lower reactivity, several case series have reported promising anti-leukemic efficacy with a low incidence of both acute and chronic GVDH [118,119,120]. Besides tolerability, a promising attempt to improve the therapeutic activity of CAR-T cells involved the enhancement of T-cell homing into the bone marrow niche by engineering CAR-T cells with overexpression of CXCR4, thus increasing the contact with residual AML cells in preclinical models [121]. Targeting the bone marrow malignant microenvironment could represent another way to bypass the resistance mechanisms of AML. With this purpose, dual CAR-T cells targeting both CD33 on myeloid leukemic cells and the mesenchymal stromal marker CD146 have been developed, and they have showed improved efficacy against AML [122]. Finally, epitope gene editing of donor hematopoietic stem cells recently appeared as a successful strategy to dissect on-target/on-tumor activity from on-target/off-tumor toxicity of immunotherapies in preclinical models of AML relapsed post HSCT, possibly representing the basis for the development of a novel, efficient compound [123].

## 11. Conclusions

The management of patients with AML relapsed post HSCT still remains a clinical challenge for both patients and physicians. Intensive treatments are often limited by severe toxicities in the early post-transplantation period. Therefore, HMAs have become, over time, the standard therapeutic approach for these patients due to the favorable safety profile. HMA can also serve as a backbone for additional anti-leukemic agents. Despite some discordant results, the addition of DLI generally appears to improve outcomes of relapsed patients with limited and manageable GvHD, even in the case of HSCT from alternative donors. The recent introduction of novel drugs in the armamentarium of AML gives the opportunity to add a third element to salvage regimens. For those patients harboring targetable mutations, IDH1/2 inhibitors Ivosidenib and Enasidenib as well as FLT3 inhibitors Sorafenib and Gilteritinib are available to combine with HMA and DLI. Conversely, patients lacking targetable mutations could benefit from the addition of Venetoclax. Whenever feasible, a second HSCT remains a valid option, especially for fit patients and for those who achieve a complete disease response with salvage regimens (Figure 2). Overall, across studies, higher response rates and longer survival are often related to pre-emptive intervention in the case of molecular relapse. Therefore, the need for close minimal residual disease monitoring after HSCT and early interventions are advisable. The future perspective appears to rely on adoptive immunotherapeutic strategies mainly represented by CAR-T cells.

## Figures and Tables

**Figure 1 ijms-24-15019-f001:**
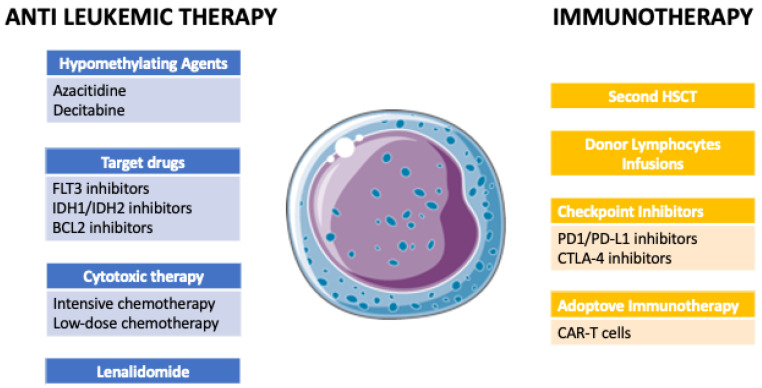
Available therapies for the treatment of acute myeloid leukemia relapsing post allogeneic stem cell transplantation.

**Figure 2 ijms-24-15019-f002:**
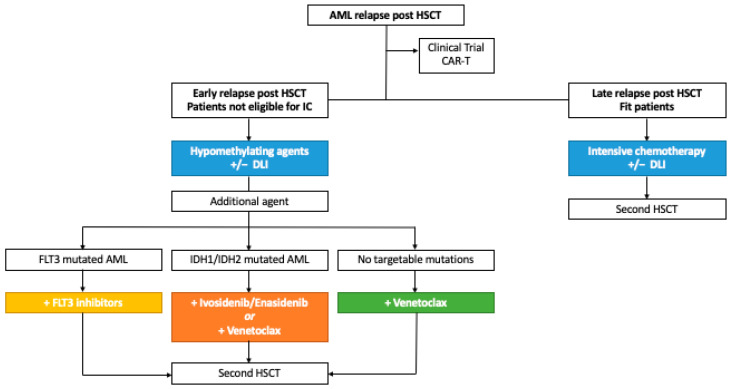
Proposed treatment algorithm for AML patients relapsed after allogeneic hematopoietic stem cell transplant. AML: acute myeloid leukemia; HSCT: allogeneic hematopoietic stem cell transplantation; IC: intensive chemotherapy; DLI: donor lymphocytes infusion.

**Table 1 ijms-24-15019-t001:** Retrospective and prospective studies investigating HMA ± DLI as treatment for post-transplantation relapse.

Study	Type of Study	Type of Relapse	HMA	Schedule	Pts (n)	DLI (n)	Median Age	MAC	MSD MUD	MMUD Haplo (n)	Time to Relapse (Days)	ORR	CR	Median Survival (Months)	2 y OS	Acute GvHD	Chronic GvHD
Platzbecker [42]	Prosp	Molec	AZA	75 mg/m^2^ ×7dd	20	0	58	10%	75%	25%	169	80%	50%	12		0%	0%
Schroeder [43]	Prosp	Morph	AZA	100 mg/m^2^ ×5dd	30	22	55	13%	13%	23%	175	47%	23%	4	17%	37%	17%
Schroeder [44]	Retr	Both	AZA	100 mg/m^2^ ×5dd –75 mg/m^2^ ×7dd	154	105	55	42%	78%	21%	185	33%	27%		29%	23%	27%
		Morph			135								21%		19%		
		Molec			19								72%		69%		
Steinmann [45]	Retr	Morph	AZA	100 mg ×3dd– 100 mg/m^2^ ×5dd	72	56	62	17%	100%				10%	3.5	54%	10%	4%
Craddock [46]	Retr	Morph	AZA	75 mg/m^2^ ×5–7dd	181	69		25%		0%	240	29%	15%		12%	7%	-
Woo [47]	Prosp	Both	AZA	75 mg/m^2^ ×7dd	39	1	52	30%	100%	0%	<100	49%	8%		25%	8%	-
		Morph			5												
		Molec			34												
Platzbecker [48]	Prosp	Molec	AZA	75 mg/m^2^ ×7dd	24	2	59				163	70%	53%		62%		
Rautenberg [49]	Retr	Both	AZA	100 mg/m^2^ ×5dd–75 mg/m^2^ ×7dd	151	105	54	36%	75%	25%	147	46%	41%			42%	26%
		Morph			92											41%	21%
		Molec			59											42%	34%
Liberatore [24]	Retr	Both	AZA	32 mg/m^2^ ×5dd –75 mg/m^2^ ×7dd	71	33	56	52%	0%	96%	270	49%	38%	7	41%	27%	18%
		Morph			40	17		45%		93%	300	38%	15%	6	19%	20%	8%
		Molec			31	16		61%		100%	240	65%	65%	15	70%	35%	19%
Poiré [50]	Prosp	Morph	AZA	35 mg/m^2^ ×5dd	49	17	60	22%	79%		146	29%	20%	6		5%	12%
Ganguly [51]	Retr	Morph	DEC	20 mg/m^2^ ×5dd	8	3	49	-	75%	25%	181	62%	38%			75%	-
Sommer [52]	Retr	Morph	DEC	20 mg/m^2^ ×5dd	26	18	59	12%	76%	24%	306	19%	15%	4.7		17%	6%
Schroeder [53]	Retr	Morph	DEC	20 mg/m^2^ ×5dd	36	22	36	36%	61%	39%	370	25%	17%		11%	19%	5%

List of abbreviations. HMAs: hypomethylating agents; Retr: retrospective; Prosp: prospective; Morph: morphological; Molec: molecular; AZA: Azacitidine; DEC: Decitabine; Pts: patients; DLIs: donor lymphocytes infusions; MAC: myeloablative conditioning regimen; MSD: matched sibling donor; MUD: matched unrelated donor; MMUD: mismatched unrelated donor; Haplo: haploidentical donor; ORR: overall response rate; CR: complete response; OS: overall survival; GvHD: graft-versus-host disease.

**Table 2 ijms-24-15019-t002:** Retrospective and prospective studies investigating targeted therapies for post-transplantation relapse.

Study	Type of Study	Salvage Treatment	Additional Drugs	Pts (n)	Prior HSCT (n)	DLI (n)	Median Age	Time to Relapse (Days)	ORR	CR	Median Survival (Months)	1 y OS	Acute GvHD	Chronic GvHD
Craddock [55]	Prosp	Lenalidomide	AZA	29	29	3	54	300	47%	20%	27 (responders)		10%	-
Schroeder [56]	Prosp	Lenalidomide	AZA	50	50	34	63	233	56%	50%	21		24%	28%
Joshi [57]	Retr	Venetoclax	AZA, DEC	29	29	0	58	270	38%	28%	2		0%	0%
Aldoss [58]	Retr	Venetoclax	AZA, DEC	33	13	0	62	-	64%	42%		53%	-	-
Byrne [59]	Retr	Venetoclax	AZA, DEC, LDAC	21	21	2	64	171	42%	38%	7.8		0%	0%
Gao [60]	Retr	Venetoclax	AZA, DEC	44	44	1	44	-	38%	34%	8.1 (responders)		-	-
Zhao [61]	Retr	Venetoclax	AZA	26	26	26	35	228	61%	27%	10		23%	-
Schuler [62]	Retr	Venetoclax	AZA, DEC	32	32	11	54	171	47%	36%	3.7		0%	3%
Chen [63]	Retr	Venetoclax	AZA, DEC	23	23	21	39	167	60%	52%	6.7		8%	8%
Zucenka [64]	Retr	Venetoclax	LDAC, ACTd	20	20	17	59	216	75%	70%	13.1		10%	-
Jin [65]	Prosp	Venetoclax	AZA, HRT	96	43	34	45	-	78%	71%	22.1		76%	18%
Metzelder [66]	Retr	Sorafenib	-	65	29	0	58	-	41%	24%	-	-	17%	-
Xuan [67]	Retr	Sorafenib	CT	83	53	58	36	153	83%	66%		47%	32%	28%
Rautenberg [68]	Retr	Sorafenib	AZA	8	8	6	43	91	75%	50%	11		50%	25%
Sid [69]	Retr	Sorafenib	AZA	5	5	0	44	280	-	60%	-	-	40%	-
De Freitas [70]	Retr	Sorafenib	AZA	13	13	1	38	93	92%	38%		22%	30%	7%
Perl [71]	Prosp	Gilteritinib	-	247	48	0	62	-	-	35%	8.3		-	-
Daver [72]	Prosp	Gilteritinib	Venetoclax	61	19	0	63	-	-	67%	8.8		-	-

List of abbreviations. Retr: retrospective; Prosp: prospective; AZA: Azacitidine, DEC: Decitabine; LDAC: low-dose cytarabine; ACTd: Actinomycin D; HRT: Homoharringtonine; CT: chemotherapy; Pts: patients; DLIs: donor lymphocytes infusions; ORR: overall response rate; CR: complete response; OS: overall survival; GvHD: graft-versus-host disease.

## Data Availability

Not applicable.

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
