# Peer review of "Novel Approaches to Treatment of Acute Myeloid Leukemia Relapse Post Allogeneic Stem Cell Transplantation"

_ijms, 2023, doi:10.3390/ijms241915019_

Round 1
Reviewer 1 Report
This article is a comprehensive review of the treatment of relapsed AML following HSCT.
The article summarized overall treatments systemically well.
The article did not include the efficacy or tolerability of leukemia antigen-targeted monoclonal antibody therapies ( e.g. anti-CD33 monoclonal antibodies, anti-CD123, anti-CD47, and so on).
it would be helpful to include them in the manuscript.
Author Response
Despite monoclonal antibodies proved to be effective in the treatment of both newly diagnosed and relapsed/refractory AML, data on their use post HSCT is very limited and only few case reports/case series ara available. Therefore, the section was not included in this review addressing treatment of AML relapse post HSCT.Reviewer 2 Report
A very well-written, comprehensive, and informative review.
The introduction is appropriate and states the scope of the article.
The bosy of the review quite complete.
Tables are informative.
Author Response
Thanks
Reviewer 3 Report
The authors reviewed the novel approaches for the treatment of AML with post allo-HSCT relapse. The manuscript is comprehensively written. The reviewer asks the author to add some information about CAR-T therapy to better understand the potential of the treatment.
【Minor】
・One of the problems of AML-CAR-T is cytopenia because myeloid antigens are often co-expressed on normal hematopoietic stem/progenitor cells. The other concern AML-CAR-T post-HSCT might be the impact of GVHD. Please include the information about cytopenia and GVHD if there are any references about these problems.
・P10 L225 70,71→[70,71]
・P12 L373 7→[7]
Author Response
the role of CAR-T cells in AML was highlighted in text
details on cytopenia as well as GvHD were added in text
Reviewer 4 Report
Generally, well-written with nice tables of results with different kinds of therapy. It should be added and noted some kind of hierarchy of the specific importance of these therapies. For example an algorithm of therapy could be useful in addition.
row 194: not rump up but ramp up
row 271: spacular trial (what does it mean specular trial?)
Author Response
a treatment algorithm was included in text as Figure 2
row 194: not rump up but ramp up --> edited in text
row 271: specular trial (what does it mean specular trial?) --> edited in text